# REUSING PREPROCESSING DATA AS AUXILIARY SUPERVISION IN CONVERSATIONAL ANALYSIS

## ABSTRACT

Conversational analysis systems are trained using noisy human labels and often require heavy preprocessing during multi-modal feature extraction. Using noisy labels in single-task learning increases the risk of over-fitting. However, auxiliary tasks could improve the performance of the primary task learning. This approach is known as Primary Multi-Task Learning (MTL). A challenge of MTL is the selection of beneficial auxiliary tasks that avoid negative transfer. In this paper, we explore how the preprocessed data used for feature engineering can be re-used as auxiliary tasks in Primary MTL, thereby promoting the productive use of data in the form of auxiliary supervision learning. Our main contributions are: (1) the identification of sixteen beneficially auxiliary tasks, (2) the method of distributing learning capacity between the primary and auxiliary tasks, and (3) the relative supervision hierarchy between the primary and auxiliary tasks. Extensive experiments on IEMOCAP and SEMAINE data validate the improvements over single-task approaches, and suggest that it may generalize across multiple primary tasks.

## 1 INTRODUCTION

The sharp increase in uses of video-conferencing creates both a need and an opportunity to better understand these conversations (Kim et al., 2019a). In post-event applications, analyzing conversations can give feedback to improve communication skills (Hoque et al., 2013; Naim et al., 2015). In real-time applications, such systems can be useful in legal trials, public speaking, e-health services, and more (Poria et al., 2019; Tanveer et al., 2015).

Analyzing conversations requires both human expertise and a lot of time, which is what many multimodal conversational analysis systems are trying to solve with automation. However, to build such systems, analysts often require a training set annotated by humans (Poria et al., 2019). The annotation process is costly, thereby limiting the amount of labeled data. Moreover, third-party annotations on emotions are often noisy. Noisy data coupled with limited labeled data increases the chance of overfitting (James et al., 2013).

From the perspective of feature engineering to analyze video-conferences, analysts often employ pre-built libraries (Baltrušaitis et al., 2016; Vokaturi, 2019) to extract multimodal features as inputs to training. This preprocessing phase is often computationally heavy, and the resulting features are only used as inputs. In this paper, we investigate how the preprocessed data can be re-used as auxiliary tasks in Primary Multi-Task Learning (MTL), thereby promoting a more productive use of data, in the form of auxiliary supervised learning. Specifically, our main contributions are (1) the identification of beneficially auxiliary tasks, (2) the method of distributing learning capacity between the primary and auxiliary tasks, and (3) the relative supervision hierarchy between the primary and auxiliary tasks. We demonstrate the value of our approach through predicting emotions on two publicly available datasets, IEMOCAP (Busso et al., 2008) and SEMAINE (McKeown et al., 2011).

## 2 RELATED WORKS AND HYPOTHESES

Multitask learning has a long history in machine learning (Caruana, 1997). In this paper, we focus on Primary MTL, a less commonly discussed subfield within MTL (Mordan et al., 2018). Primary

MTL is concerned with the performance on one (primary) task – the sole motivation of adding auxiliary tasks is to improve the primary task performance.

In recent years, primary MTL has been gaining attention in computer vision (Yoo et al., 2018; Fariha, 2016; Yang et al., 2018; Mordan et al., 2018; Sadoughi & Busso, 2018), speech recognition (Krishna et al., 2018; Chen & Mak, 2015; Tao & Busso, 2020; Bell et al., 2016; Chen et al., 2014), and natural language processing (NLP) (Arora et al., 2019; Yousif et al., 2018; Zalmout & Habash, 2019; Yang et al., 2019; Du et al., 2017). The benefit of adding multiple tasks is to provide inductive bias through multiple noisy supervision (Caruana, 1997; Lipton et al., 2015; Ghosn & Bengio, 1997). On the other hand, the drawback of adding multiple tasks increases the risk of negative transfer (Torrey & Shavlik, 2010; Lee et al., 2016; 2018; Liu et al., 2019; Simm et al., 2014), which leads to many design considerations. Two of such considerations are, identifying (a) what tasks are beneficial and (b) how much of the model parameters to share between the primary and auxiliary tasks. In addition, because we are performing Primary MTL, we have the third consideration of (c) whether we should prioritize primary supervision by giving it a higher hierarchy than the auxiliary supervision.

In contrast with previous MTL works, our approach (a) identifies sixteen beneficially auxiliary targets, (b) dedicates a primary-specific branch within the network, and (c) investigates the efficacy and generalization of prioritizing primary supervision across eight primary tasks.

Since our input representation is fully text-based, we dive deeper into primary MTL in the NLP community. Regarding model architecture designs for primary MTL in NLP, Søgaard & Goldberg (2016) found that lower-level tasks like part-of-speech tagging, are better kept at the lower layers, enabling the higher-level tasks like Combinatory Categorical Grammar tagging to use these lower-level representations. In our approach, our model hierarchy is not based on the difficulty of the tasks, but more simply, we prioritize the primary task. Regarding identifying auxiliary supervisors in NLP, existing works have included tagging the input text (Zalmout & Habash, 2019; Yang et al., 2019; Søgaard & Goldberg, 2016). Text classification with auxiliary supervisors have included research article classification (Du et al., 2017; Yousif et al., 2018), and tweet classification (Arora et al., 2019). There is a large body of work in multimodal sentiment analysis, but not in the use of multimodal auxiliary supervisors, as detailed in the next paragraph.

Multimodal analysis of conversations has been gaining attention in deep learning research, particularly for emotion recognition in conversations (Poria et al., 2019). The methods in the recent three years have been intelligently fusing numeric vectors from the text, audio, and video modalities before feeding it to downstream layers. This approach is seen in MFN (Zadeh et al., 2018a), MARN (Zadeh et al., 2018b), CMN (Hazarika et al., 2018b), ICON (Hazarika et al., 2018a), DialogueRNN (Majumder et al., 2019), and M3ER (Mittal et al., 2020). Our approach is different in two ways. (1) Our audio and video information is encoded within text before feeding only the text as input. Having only text as input has the benefits of interpretability, and the ability to present the conversational analysis on paper (Kim et al., 2019b). This is similar to how the linguistics community performs manual conversational analysis using the Jefferson transcription system (Jefferson, 2004), where the transcripts are marked up with symbols indicating how the speech was articulated. (2) Instead of using the audio and video information as only inputs to a Single Task Learning (STL) model, the contribution of this paper is that we demonstrate how to use multimodal information in both input and as auxiliary supervisors to provide inductive bias that helps the primary task.

**Hypothesis H1: The introduced set of auxiliary supervision features improves primary MTL.**
We introduce and motivate the full set of sixteen auxiliary supervisions, all based on existing literature: these are grouped into four families, each with four auxiliary targets. The four families are (1) facial action units, (2) prosody, (3) historical labels, and (4) future labels:
(1) Facial action units, from the facial action coding system identifies universal facial expressions of emotions (Ekman, 1997). Particularly, AU 05, 17, 20, 25 have been shown to be useful in detecting depression (Yang et al., 2016a; Kim et al., 2019b) and rapport-building (Anonymous, 2021).
(2) Prosody, the tone of voice – happiness, sadness, anger, and fear – can project warmth and attitudes (Hall et al., 2009), and has been used as inputs in emotions detection (Garcia-Garcia et al., 2017).
(3 and 4) Using features at different historical time-points is a common practice in statistical learning, especially in time-series modelling (Christ et al., 2018). Lastly, predicting future labels as auxiliary tasks can help in learning (Caruana et al., 1996; Cooper et al., 2005; Trinh et al., 2018; Zhu et al., 2020; Shen et al., 2020). Inspired by their work, we propose using historical and future

(up to four talkturns ago or later) target labels as auxiliary targets. Although historical and future target labels can be used as a pre-training objective and fine-tuned on the current target label, sequential transfer learning is not the focus of this paper.

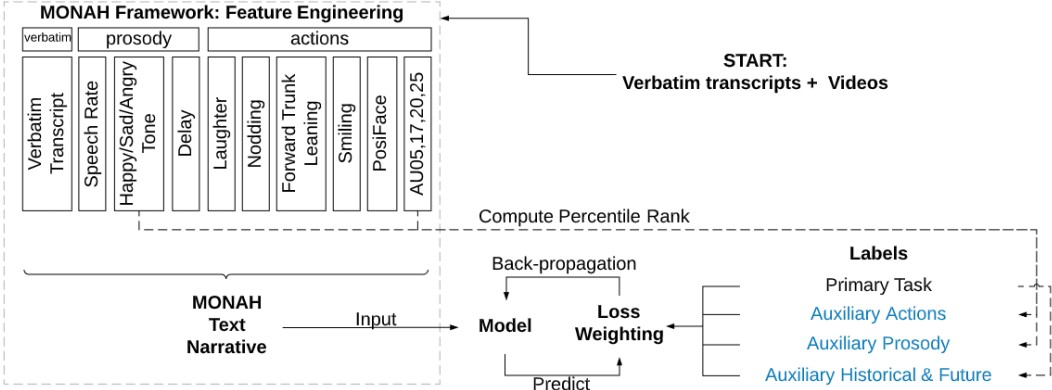

Figure 1: Reusing components (dotted lines) of the feature engineering process as auxiliary targets (in blue). The MONAH framework (Anonymous, 2021) is introduced in section 4.2

Given that we are extracting actions and prosody families as inputs, we propose to explore whether they can be reused as supervisors (see Fig. 1). Our hypothesis **H1** is that re-using them as auxiliary supervision improves primary MTL. This is related to using hints in the existing MTL literature, where the auxiliary tasks promote the learning of the feature (Cheng et al., 2015; Yu & Jiang, 2016).

**Hypothesis H2: When the primary branch is given maximum learning capacity, it would not be outperformed by models with primary branch having less than the maximum learning capacity.**
Deeper models with higher learning capacity produce better results (Huang et al., 2019; Nakkiran et al., 2019; Menegola et al., 2017; Blumberg et al., 2018; Romero et al., 2014). Also, since the auxiliary branch is shared with the primary supervision, the auxiliary capacity should be limited to enforce information transfer and improve performance (Wu et al., 2020). Therefore, given a fixed learning capacity budget, our hypothesis **H2** implies that we should allocate the maximum learning capacity to the primary branch because we care only about the primary task performance.

**Hypothesis H3: Auxiliary supervision at the lower hierarchy yields better primary MTL as compared to flat-MTL.**
Having the auxiliary tasks at the same supervisory level as the primary task is inherently sub-optimal because we care only about the performance of the primary task (Mordan et al., 2018). To prioritize the primary task, we could change the model architecture such that the auxiliary supervision is at the lower hierarchy than the primary supervision, which will be discussed in the next section.

## 3 MODEL ARCHITECTURE

### 3.1 FLAT-MTL HIERARCHICAL ATTENTION MODEL

We start with an introduction of the Hierarchical Attention Model (HAN) (Yang et al., 2016b). We chose HAN because of its easy interpretability as it only uses single-head attention layers. There are four parts to the HAN model, (1) text input, (2) word encoder, (3) talkturn encoder, and (4) the predictor. In our application, we perform our predictions at the talkturn-level for both IEMOCAP and SEMAINE. For notation, let $s_i$ represent the $i$-th talkturn and $w_{it}$ represent the $t$-th word in the $i$-th talkturn. Each single talkturn can contain up to $T$ words, and each input talkturn can contain up to $L$ past talkturns to give content context (to discuss in section 4.2).

**Text Input** Given a talkturn of words, we first convert the words into vectors through an embedding matrix $W_e$, and the word selection one-hot vector, $w_{it}$.

**Word encoder** The word encoder comprises of bidirectional GRUs (Bahdanau et al., 2014) and a single head attention to aggregate word embeddings into talkturn embeddings. Given the vectors $x_{it}$, the bidirectional GRU reads the words from left to right as well as from right to left (as indicated by the direction of the GRU arrows) and concatenates the two hidden states together to form $h_{it}$. We then aggregate the hidden states into one talkturn embedding through the attention mechanism. $u_{it}$ is the hidden state from feeding $h_{it}$ into a one-layer perceptron (with weights $W_w$ and biases $b_w$). The attention weight ($\alpha_{it}$) given to $u_{it}$ is the softmax normalized weight of the similarity between itself ($u_{it}$) and $u_w$, which are all randomly initialized and learnt jointly.

$$x_{it} = W_e w_{it}, t \in [1, T] \qquad\qquad u_{it} = relu(W_w h_{it} + b_w)$$

$$\overrightarrow{h}_{it} = \overrightarrow{GRU}(x_{it}), t \in [1, T] \qquad\qquad \alpha_{it} = \frac{exp(u_{it}^\top u_w)}{\Sigma_t exp(u_{it}^\top u_w)}$$

$$\overleftarrow{h}_{it} = \overleftarrow{GRU}(x_{it}), t \in [T, 1] \qquad\qquad s_i = \Sigma_t \alpha_{it} u_{it}$$

$$h_{it} = (\overrightarrow{h}_{it}, \overleftarrow{h}_{it})$$

**Talkturn encoder** With the current and past talkturn embeddings (content context, to discuss in section 4.2), the talkturn encoder aggregates them into a single talkturn representation ($v$) in a similar fashion, as shown below.

$$\overrightarrow{h}_i = \overrightarrow{GRU}(s_i), i \in [1, L] \qquad\qquad u_i = relu(W_s h_i + b_s)$$

$$\overleftarrow{h}_i = \overleftarrow{GRU}(s_i), i \in [L, 1] \qquad\qquad \alpha_i = \frac{exp(u_i^\top u_s)}{\Sigma_i exp(u_i^\top u_s)}$$

$$h_i = (\overrightarrow{h}_i, \overleftarrow{h}_i) \qquad\qquad v = \Sigma_i \alpha_i u_i$$

The simplest way of adding the sixteen (four auxiliary targets from four families of auxiliary supervision) auxiliary task predictors would be to append them to where the primary task predictor is, as illustrated in Fig. 2. That way, all predictors use the same representation $v$. We refer to this architecture as flat-MTL because the auxiliary supervision is at the same level as the primary supervision. We are unable to test **H2** and **H3** using this architecture.

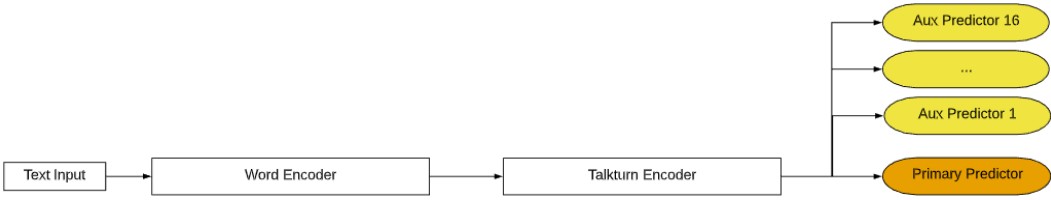

Figure 2: Forward pass of the Flat-MTL HAN architecture. Auxiliary tasks (yellow) are added at the same level of the primary task (orange).

## 3.2 HAN-ROCK

We adapted[1] the ROCK architecture (Mordan et al., 2018) which was built for Convolutional Neural Networks (LeCun et al., 1995) found in ResNet-SSD (He et al., 2016; Liu et al., 2016) to suit GRUs (Bahdanau et al., 2014) found in HAN (Yang et al., 2016b) (see Fig. 3).

To study **H3**, we bring the auxiliary task predictors forward, so that the auxiliary supervision is of a lower hierarchy than primary supervision (see Fig. 3). It is of a lower hierarchy because the back-propagation from the primary supervision is able to temper the back-propagation from the auxiliary

---

[1]Implementation available at https://github.com/anonymous/placeholder; Please see attached supplementary material for implementation details during the review phase.

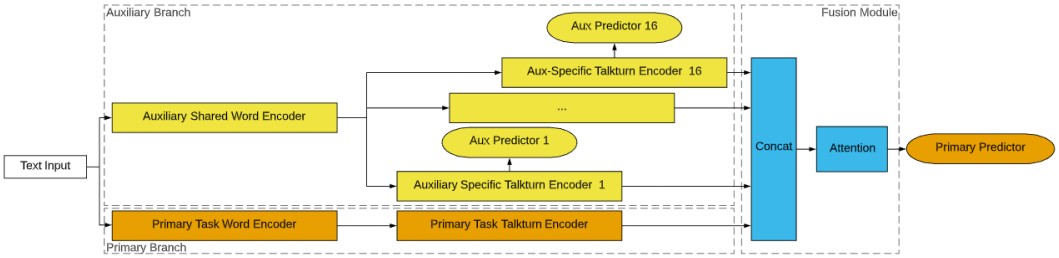

Figure 3: Forward pass of the HAN-ROCK architecture. There is a primary branch (orange) where auxiliary supervision (yellow) can not influence. The fusion module (blue) aggregates the talkturn embeddings from all tasks into one.

supervision but not vice-versa. This also sets us up to study **H2**, the impact of distributing different learning capacities to the auxiliary and primary branch. Each of the auxiliary tasks has its own talkturn encoder but shares one word encoder in the auxiliary branch (to keep the network small). Subscript $a$ indicates whether the word encoder is for the primary or auxiliary branch:

$$x_{it} = W_e w_{it}, t \in [1, T] \qquad\qquad u_{ait} = relu(W_{aw}h_{ait} + b_{aw})$$

$$\overrightarrow{h}_{ait} = \overrightarrow{GRU}_a(x_{it}), t \in [1, T], a \in \{pri, aux\} \qquad \alpha_{ait} = \frac{exp(u_{ait}^\top u_{aw})}{\Sigma_t exp(u_{ait}^\top u_{aw})}$$

$$\overleftarrow{h}_{ait} = \overleftarrow{GRU}_a(x_{it}), t \in [T, 1], a \in \{pri, aux\} \qquad s_{ai} = \Sigma_t \alpha_{ait} u_{ait}$$

$$h_{ait} = (\overrightarrow{h}_{ait}, \overleftarrow{h}_{ait})$$

Each task has its own talkturn encoder. Subscript $b$ indicates which of the seventeen tasks – the primary talkturn task or one of the sixteen auxiliary tasks – is the talkturn encoder is dedicated to:

$$\overrightarrow{h}_{abi} = \overrightarrow{GRU}_{ab}(s_{ai}), i \in [1, L], a \in \{pri, aux\}, b \in \{pri, aux1, aux2, ..., aux16\}$$

$$\overleftarrow{h}_{abi} = \overleftarrow{GRU}_{ab}(s_{ai}), i \in [L, 1], a \in \{pri, aux\}, b \in \{pri, aux1, aux2, ..., aux16\}$$

$$h_{abi} = (\overrightarrow{h}_{abi}, \overleftarrow{h}_{abi}) \qquad\qquad \alpha_{abi} = \frac{exp(u_{abi}^\top u_b)}{\Sigma_i exp(u_{abi}^\top u_b)}$$

$$u_{abi} = relu(W_b h_{abi} + b_b) \qquad\qquad v_{ab} = \Sigma_i \alpha_{abi} u_{abi}$$

The seventeen talkturn embeddings ($v_{ab}$) goes through a concatenation, then the single head attention, aggregating talkturn embeddings across seventeen tasks into one talkturn embedding for the primary task predictor. Subscript $c$ pertains to the fusion module.

$$\text{concatenation: } v_c = (v_{ab}), a \in \{pri, aux\}, b \in \{pri, aux1, aux2, ..., aux16\}$$

$$\text{attention: } \alpha_c = \frac{exp(v_c^\top u_c)}{\Sigma_c exp(v_c^\top u_c)}$$

$$\text{overall primary talkturn vector: } v = \Sigma_c \alpha_c v_c$$

## 4 EXPERIMENTS

### 4.1 DATA AND PRIMARY TASKS

We validate our approach using two datasets with a total of eight primary tasks: the IEMOCAP (Busso et al., 2008) and the SEMAINE (McKeown et al., 2011) datasets. Both datasets are used

in multimodal emotions detection research (Poria et al., 2019). We divided the datasets into train, development, and test sets in an approximate 60/20/20 ratio such that the sets do not share any speaker (Appendix A.1 details the splits).

The target labels of the eight primary tasks are all at the talkturn-level. The four primary tasks of IEMOCAP consists of the four-class emotions classification (angry, happy, neutral, sad), and three regression problems – valence (1-negative, 5-positive), activation (1-calm, 5-excited), and dominance (1-weak, 5-strong). The four-class emotions classification target is common for IEMOCAP (Latif et al., 2020; Xia & Liu, 2015; Li et al., 2019; Hazarika et al., 2018b; Mittal et al., 2020), albeit not universal. Some researchers have gone up to five (Chang & Scherer, 2017) or six (Majumder et al., 2019; Hazarika et al., 2018a) or nine-class emotions classification (Zadeh et al., 2018a) target.

For SEMAINE, there are four regression problems – activation, intensity, power, valence. We note that the valence, power, and activation tasks might be related across the two datasets, but cross-domain learning is beyond the scope of this paper. We use two standard evaluation metrics, mean absolute error (MAE), and 4-class weighted mean classification accuracy, MA(4).

## 4.2 INPUT

Multimodal feature extraction is computed using the MONAH framework (Anonymous, 2021). This framework uses a variety of pre-trained models to extract nine multimodal features, associated with the prosody of the speech and the actions of the speaker, and weaves them into a multimodal text narrative. We refer the reader to Anonymous (2021) for the details and efficacy of the MONAH framework. The benefit of the created narrative is that it describes what is said together with how it is said for each talkturn, giving richer nonverbal context to the talkturn (see Fig. 4 for an example). Being fully text-based means that the analysis product can be printed out on paper, without the need for speakers nor monitors to replay the conversation on a computer.

| Verbatim: | "The woman said no." |
|---|---|
| MONAH: | "The woman raised chin. The woman sadly and slowly said no." |

Figure 4: Example of a MONAH transcript.

In addition to nonverbal context, we concatenated a variable number of preceding talkturns to the current talkturn as content context. Content context has been proven to be useful in CMN (Hazarika et al., 2018b) and ICON (Hazarika et al., 2018a), DialogueRNN (Majumder et al., 2019). The content-context size is tuned as a hyperparameter. The resulting multimodal text narrative, consisting of both nonverbal and context context, is used as the sole input to the model. The impact of progressively removing nonverbal and content context is not a contribution of this paper and hence not analyzed.

## 4.3 AUXILIARY TARGETS

We first clarify the method of extraction for the auxiliary families. The OpenFace algorithm (Baltrušaitis et al., 2016) is used to extract the four continuous facial action units (AU) – AU 05, 17, 20, 25. The Vokaturi algorithm (Vokaturi, 2019) is used to extract the four continuous dimensions in the tone of voice – happiness, sadness, anger, and fear. As for historical and future features, we simply look up the target label for the past four talkturns, and future four talkturns. If any of the historical or future labels are not available, for example, the target label four talkturns ago is not available for the third talkturn, we substitute with the next nearest non-missing label.

For all auxiliary targets that reused the input features (actions and prosody), we converted them into a percentile rank that has the range [0,1] using the values from the train partition. This is a subtle but note-worthy transformation. When reusing an input as an auxiliary target, it would be trivial if the input can easily predict the target. For example, given the following MONAH transcript as input, "The woman sadly and slowly said no." It would be trivial to use a binary (quantized) auxiliary target of "was the tone sad?" because we would only be training the model to look for the word "sadly". However, if the auxiliary target is a percentile rank (less quantized) of the sadness in tone, then the presence of the word "sadly" increases the predicted rank, but the model could still use the rest of the nonverbal cues ("slowly") and what is being said ("no") to predict the degree of sadness.

That way, representations learnt for the auxiliary tasks uses more of the input. Additionally, this transformation also helped us side-step the decision to use the multimodal streams *exclusively* in the inputs or in the supervision (Caruana & De Sa, 1997; Caruana & de Sa, 1998) because we could use them in both.

Percentile rank also has the convenient property of having the range [0,1]. We scaled the percentile ranks so that they all have the same range as the primary task (see appendix A.2 for transformation details). This to ensures that if we assigned equal loss weights to all tasks, the contribution of every task (both auxiliary and primary) is of the same order of magnitude, an important consideration for MTL (Gong et al., 2019; Hassani & Haley, 2019; Sener & Koltun, 2018).

### 4.4 MODELS, TRAINING, AND HYPERPARAMETERS TUNING

The overall loss is calculated as the weighted average across all seventeen tasks: (1) we picked a random weight for the primary task from the range [0.50, 0.99]; this ensures that the primary task has the majority weight. (2) For the remaining weights (1 - primary weight), we allocated them to the sixteen auxiliary tasks by: (a) random, (b) linearly-normalized mutual information, or (c) softmax-normalized mutual information. (a) is self-explanatory. As for (b) and (c), mutual information has been shown that it is the best predictor – compared to entropy and conditional entropy – of whether the auxiliary task would be helpful to primary MTL (Bjerva, 2017). We computed the mutual information (vector $m$) of each auxiliary variable with the primary target variable (Kraskov et al., 2004; Ross, 2014) using scikit-learn (Pedregosa et al., 2011). Then, we linearly-normalized or softmax-normalized $m$ to sum up to 1. Finally, we multiplied the normalized $m$ with the remaining weights from (2); this ensures that the primary weight and the sixteen auxiliary weight sum up to one. (a), (b), and (c) have ten trials each during hyper-parameters tuning.

Two variants of the HAN architectures are used (Fig 2 and 3). Glove word embeddings (300-dimensions) are used to represent the words (Pennington et al., 2014). Hyper-parameters tuning is crucial because different combinations of primary and auxiliary tasks require different sets of hyper-parameters. For hyperparameters tuning, we used random search (Bergstra & Bengio, 2012) with thirty trials. We tuned the learning rate, batch size, L2 regularization, the number of GRUs assigned to the primary and auxiliary branches, the auxiliary weights assignment, the content-context size, and lastly the GRU dropout and recurrent dropout (as detailed in Appendix A.3).

Training is done on a RTX2070 or a V100 GPU, for up to 350 epochs. Early stopping is possible via the median-stopping rule (Golovin et al., 2017) after the fifth epoch and after every two epochs (i.e., at epoch number 5, 7, 9, ..., 349). Appendix A.4 details the hyperparameters of models that performed the best on the development set.

For hypotheses testing, we bootstrapped confidence intervals of the test set performance of both baseline and challenger models as well as the confidence intervals of the differences in performances. Please see appendix A.5 for details on the bootstrap procedure.

## 5 RESULTS AND DISCUSSION

The key takeaways are: (**H1**) The introduced set of auxiliary supervision improves primary MTL significantly in six of the eight primary tasks. (**H2**) Maximum learning capacity should be given to the primary branch as a default. (**H3**) HAN-ROCK is unlikely (in one of the eight tasks) to degrade primary MTL significantly, and sometimes significantly improves it (in four of the eight tasks).

(**H1**): To test **H1** (whether the introduced set of auxiliary supervision improves Primary MTL), we first train the model with all sixteen auxiliary targets (from families actions, prosody, historical, and future). Then, to differentiate the effect from the historical and future supervision, we set the loss weights from historical and future targets to be zero; effectively, there is only supervision from eight auxiliary targets (actions and prosody). Lastly, for the baseline model (no auxiliary supervision), we set the loss weights from all sixteen auxiliary targets to zero.

Given auxiliary supervision, the model significantly outperforms the baseline of not having auxiliary supervision in six out of the eight primary tasks (Table 1). Comparing the baseline model with the model with two auxiliary target families, they significantly outperformed the baseline model in five out of eight primary tasks. The addition of two auxiliary target families (historical and

Table 1: **H1** Results. *: the model performance has a statistically significant difference with the baseline model. a: action, p: prosody, h: historical labels, f: future labels.

| | IEMOCAP | | | | SEMAINE | | | |
|---|---|---|---|---|---|---|---|---|
| Aux. Target | Classif. MA(4) | Val. MAE | Act. MAE | Dom. MAE | Act. MAE | Int. MAE | Pow. MAE | Val. MAE |
| None (Baseline) | 0.625 | 0.527 | 0.518 | 0.667 | 0.194 | 0.238 | 0.170 | 0.177 |
| ap | 0.715* | 0.538 | 0.507 | 0.600* | 0.184* | 0.218* | 0.167* | 0.178 |
| aphf | 0.706* | 0.497* | 0.504 | 0.587* | 0.187 | 0.231 | 0.165* | 0.169 |

future labels) sometimes significantly improved primary MTL (valence in IEMOCAP), but it also sometimes significantly made it worse (activation and intensity in SEMAINE). This shows that the value of auxiliary tasks, and the associated risk of negative transfer, depends on the auxiliary task.

(**H2**): To test **H2** (whether maximum learning capacity should be given to the primary branch), we let $P$ represent the number of GRU assigned to the primary talkturn encoder, and $A$ represent the number of GRU assigned to each of the sixteen auxiliary talkturn encoder. We constrained $P + A$ to be equal to 257. During our experiments, we set $P$ to 1, 64, 128, 192, and 256. We set 256 as the baseline model because it is the maximum learning capacity we can give to the primary branch while giving 1 GRU ($= 257 - 256$) to each of the sixteen auxiliary talkturn encoders.

Table 2: **H2** Results. *: the model performance has a statistically significant difference with the baseline model ($P = 256$). ˆ: Assigning 1 GRU to the auxiliary task talkturn encoder yields a statistically significant difference with assigning 0 GRU.

| | IEMOCAP | | | | SEMAINE | | | |
|---|---|---|---|---|---|---|---|---|
| Pri. GRU | Classif. MA(4) | Val. MAE | Act. MAE | Dom. MAE | Act. MAE | Int. MAE | Pow. MAE | Val. MAE |
| 256 (Baseline) | 0.715ˆ | 0.538 | 0.507 | 0.587ˆ | 0.187 | 0.231 | 0.165ˆ | 0.169ˆ |
| 192 | 0.736 | 0.509 | 0.518 | 0.604 | 0.187 | 0.228 | 0.165 | 0.184* |
| 128 | 0.711 | 0.537 | 0.512 | 0.597 | 0.189 | 0.216 | 0.176* | 0.196* |
| 64 | 0.687 | 0.540 | 0.507 | 0.593 | 0.191 | 0.234 | 0.167 | 0.192* |
| 1 | 0.656 | 0.554 | 0.509 | 0.599 | 0.190 | 0.229 | 0.168* | 0.191* |

In all primary tasks, the baseline model of assigning 256 GRUs to the primary branch is not significantly outperformed by models that assigned 1, 64, 128, 192 GRUs (Table 2). Generally, the performance decreased as the number of GRUs assigned to the primary talkturn encoder decreased from 256 to 1. We observed significantly worse performance in two out of eight tasks – in power and valence in SEMAINE. Also, assigning 256 GRU to the primary talkturn encoders and 1 to each of the sixteen auxiliary talkturn encoders yields the smallest model[2], and thus trains the fastest. Therefore, we recommend that the maximum capacity be given to the primary branch as a default.

That said, the presence of an auxiliary branch is still important. The baseline of **H1** (no auxiliary supervision, Table 1) can be approximated as $P=256 + 16 \times 1$, $A=0$ because the model architecture is the same, except that the loss weights of all auxiliary tasks are zero. We compared the former to the baseline in Table 2, and found that four out of eight primary tasks have significant improvements by changing the number of talkturn encoders assigned to each auxiliary task from zero to *one*.

(**H3**): To test **H3** (whether auxiliary supervision should be given a lower hierarchy), we compare the results from the flat-MTL HAN architecture (baseline) against the HAN-ROCK architecture (Table 3). Placing auxiliary supervision at the lower hierarchy significantly improves primary MTL in four out of eight tasks. In only one out of eight tasks (power in SEMAINE), auxiliary supervision significantly degrades primary MTL. We posit that further improvements are possible through the fusion module with future research.

---

[2]As opposed to, say, assigning 1 GRU to the primary talkturn coder and 256 GRU to each of the sixteen auxiliary talkturn encoder.

Table 3: **H3** Results. *: the model performance has a significant difference with the baseline.

| | IEMOCAP | | | | SEMAINE | | | |
|---|---|---|---|---|---|---|---|---|
| **Hierarchy** | **Classif. MA(4)** | **Val. MAE** | **Act. MAE** | **Dom. MAE** | **Act. MAE** | **Int. MAE** | **Pow. MAE** | **Val. MAE** |
| Flat (Baseline) | 0.699 | 0.520 | 0.526 | 0.606 | 0.183 | 0.230 | 0.164 | 0.185 |
| HAN-ROCK | 0.715 | 0.538 | 0.507* | 0.600* | 0.184 | 0.218* | 0.167* | 0.178* |

## 5.1 CLASS-WISE PERFORMANCE AND STATE-OF-THE-ART

We discuss the IEMOCAP classification model in depth by investigating the class-wise performance under the three hypotheses (Table 4). Generally, we found that all hypotheses effects are stronger in lower resource labels (sad and anger). We also present the performance of M3ER (Mittal et al., 2020), a previous state-of-the-art (SoTA) approach. We do not expect the performance of our text-only input to match the SoTA approach, which is confirmed in Table 4. By fusing numerical vectors from the three modalities prevalent in SoTA approaches (Zadeh et al., 2018a;b; Hazarika et al., 2018b;a; Majumder et al., 2019; Mittal et al., 2020), the inputs are of a much higher granularity as compared to our approach of describing the multimodal cues using discrete words. Although the text-based input is likely to constrain model performance, the multimodal transcription could be helpful for a human to analyze the conversation even before supervised learning. We could also overlay the model perspective on the multimodal transcription to augment human analysis (see Appendix A.6).

Table 4: Class-wise classification F1 score on IEMOCAP. Baseline (challenger) refers to HAN-Rock architecture under the three hypotheses. *: the challenger performance has a statistically significant difference with the baseline model.

| Distribution | | H1 | | H2 | | H3 | | SoTA |
|---|---|---|---|---|---|---|---|---|
| **Label** | **Count** | **Baseline Aux target: None** | **Challenger Aux target: aphf** | **Baseline Pri GRU: 256** | **Challenger Pri GRU: 1** | **Baseline Hierarchy: Flat** | **Challenger Hierarchy: HAN ROCK** | **M3-ER** |
| Sad | 1,084 | 0.573 | 0.689* | 0.699 | 0.591* | 0.674 | 0.704* | 0.775 |
| Anger | 1,103 | 0.531 | 0.683 | 0.752 | 0.672* | 0.657 | 0.720* | 0.862 |
| Happy | 1,636 | 0.772 | 0.784 | 0.776 | 0.754* | 0.804 | 0.806 | 0.862 |
| Neutral | 1,708 | 0.664 | 0.636 | 0.688 | 0.627* | 0.645 | 0.631 | 0.745 |

## 6 CONCLUSION, LIMITATIONS, AND FUTURE WORK

We proposed to re-use feature engineering pre-processing data as auxiliary tasks in primary MTL. Three hypotheses were tested for primary MTL. The experimental results confirm **H1** – Introducing our set of sixteen auxiliary supervisors resulted in better performance in most primary tasks. For **H2**, maximum learning capacity should be given to the primary branch. Lastly, for **H3**, placing the auxiliary supervision in a lower hierarchy is unlikely to hurt performance significantly, and it sometimes significantly improves performance. This is encouraging news for multi-modal conversational analysis systems as we have demonstrated how pre-processed data can be used twice to improve performance, once as inputs, and again as auxiliary tasks. This paper has limitations. The first limitation is that the solutions are evaluated on eight tasks in the conversational analysis domain, and it is not clear if these would generalize outside of this domain. The second limitation is that we have evaluated on HAN, but not other network architectures.

A challenge to be addressed is the apriori selection of the auxiliary targets. Future research could investigate targets selection, including how to use a much larger range of auxiliary targets, how to decide the optimum number of auxiliary targets, and whether it is possible to perform these automatically.

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

# A   APPENDIX

## A.1   DATASET PARTITIONS

We detail the dataset partition in 5 for aid reproducibility.

Table 5: Dataset partitions

| IEMOCAP | | | SEMAINE | | |
|---------|---------|-------|---------|---------|-------|
| **Partition** | **Session** | **Count** | **Partition** | **Session** | **Count** |
| Train | Ses01F | 861 | Train | 2008.12.05.16.03.15 | 424 |
| Train | Ses01M | 958 | Train | 2009.01.30.12.00.35 | 404 |
| Train | Ses02F | 889 | Train | 2009.02.12.10.49.45 | 686 |
| Train | Ses02M | 922 | Train | 2009.05.15.15.04.29 | 436 |
| Train | Ses03F | 958 | Train | 2009.05.29.14.30.05 | 760 |
| Train | Ses03M | 1178 | Train | 2009.05.22.15.17.45 | 668 |
| Dev | Ses04F | 1105 | Train | 2009.05.25.11.23.09 | 928 |
| Dev | Ses04M | 998 | Train | 2009.05.26.10.19.53 | 790 |
| Test | Ses05F | 1128 | Train | 2009.06.05.10.14.28 | 732 |
| Test | Ses05M | 1042 | Train | 2009.06.15.12.13.06 | 958 |
| | | | Train | 2009.06.19.14.01.24 | 586 |
| | | | Train | 2009.10.27.16.17.38 | 440 |
| | | | Dev | 2008.12.19.11.03.11 | 188 |
| | | | Dev | 2009.01.06.14.53.49 | 472 |
| | | | Dev | 2009.05.12.15.02.01 | 448 |
| | | | Dev | 2009.05.08.11.28.48 | 752 |
| | | | Dev | 2009.06.26.14.38.17 | 404 |
| | | | Test | 2008.12.14.14.47.07 | 372 |
| | | | Test | 2009.01.06.12.41.42 | 264 |
| | | | Test | 2009.01.28.15.35.20 | 364 |
| | | | Test | 2009.06.26.14.38.17 | 440 |
| | | | Test | 2009.06.26.14.09.45 | 448 |

## A.2   SCALING THE AUXILIARY TARGETS

We detail the operations in scaling the percentile scores that range [0,1] to various primary tasks. For IEMOCAP Primary Tasks that are regression problems, we multiply the percentile score by 4 and add 1 to obtain the range [1,5]. For IEMOCAP classification task, we leave the auxiliary targets in the range of [0,1]. As for SEMAINE tasks, which are all regression problems, we multiply the percentile score by 2 and minus 1 to obtain the range [-1,1].

## A.3   RANGE OF HYPER PARAMETERS TUNED

We document the range of hyper parameters tuned in Table 6 to aid reproducibility.

## A.4   HYPER PARAMETERS WITH BEST DEVELOPMENT SET PERFORMANCE

We document our best performing hyper parameters in Table 7 to aid reproducibility.

## A.5   DETAILS OF COMPUTING THE BOOTSTRAP CONFIDENCE INTERVAL

Baseline models for each hypothesis are detailed in section 2. All non-baseline models are referred to as challenger models. We created 1000 bootstrap samples of the test set performance by (1) resampling the development set performance, then (2) selecting the set of hyperparameters that resulted in the best development set performance, and (3) looking up the test set performance given the set of best-performing hyperparameters for the development set. To judge whether the challenger outperforms the baseline, we computed the 95 percent confidence interval by (1) performing

Table 6: Range of Hyperparameters tuned. U: Uniform sampling, LU: Uniform sampling on the logarithmic scale.

| Name | Min. | Max | Sampling |
|---|---|---|---|
| Learning rate | $2^{-10}$ | $2^{-5}$ | LU |
| Batch-Size | 32 | 256 | U |
| Pri GRU | 1, 64, 128, 192, 256 | | U |
| Aux GRU | 257 - (minus) Pri GRU | | |
| Loss-weights Assignment | Random, Linear-Normalized, Softmax-Normalized | | U |
| L2 Regularization | 0.0 | 0.50 | LU |
| Content-Size | 1 | 30 | U |
| GRU dropout | 0.01 | 0.50 | U |
| Recurrent dropout | 0.01 | 0.50 | U |

element-wise subtraction between the resampled test set performance of the baseline against the challenger, (2) removing the top and bottom 2.5 percent from the differences, and (3) observing whether the remaining 95 percent confidence interval includes zero. If it does not include zero, then the difference is statistically significant.

## A.6 VISUALIZATION FROM HAN-ROCK

We demonstrate how the HAN-ROCK model could be used to support humans analyze conversations using only text-based inputs. We visualized the attention weights from two models, (1) MTL refers to the classification model with auxiliary supervisors, whilst (2) STL refers to the same model architecture, but its auxiliary supervisors' loss weights are set to zero. In principle, the MTL model should exhibit attention weights that are less likely to overfit because the weights are tempered by auxiliary supervisors. We observe that across the two models, both use the historical talkturns more so than the current talkturn; secondly, both assign high attention to the second word of the talkturns, which is interesting because the second word is where the multimodal annotations are inserted.

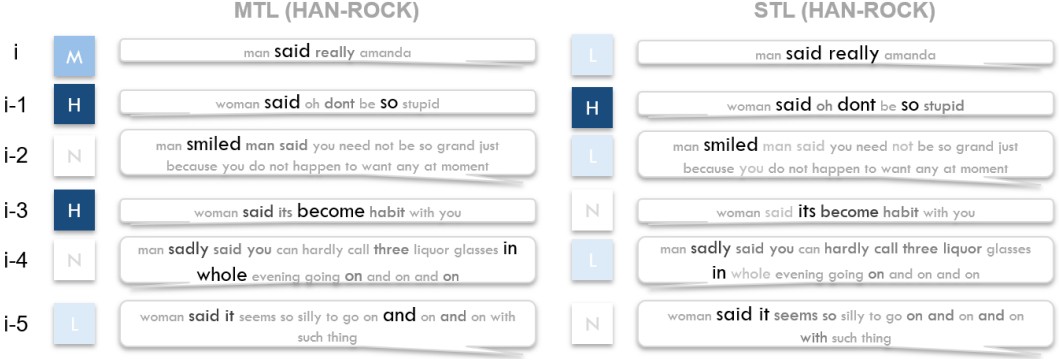

Figure 5: Conversation analysis example. Both models predicted the class label (anger) correctly. The left-most column denotes the talkturn context – $i$ refers to the current talkturn where the target emotion class is predicted. The square boxes indicate the level of attention (N: None, L: Low, M: Medium, H: High) assigned to the talkturn. Within the talkturn, we also enlarge and darken the font color ti visualize higher attention weights.

As explained in Section 3.2, there are three levels of attention over the internal representations, word ($\alpha_{it}$), talkturn ($\alpha_{abi}$), and task ($\alpha_c$). To compute the overall word and talkturn attention, we compute the weighted average of $\alpha_{it}$ and $\alpha_{abi}$ using the $\alpha_c$ (task attention) as weights. Once we have the overall word and talkturn attention, we standardize the weights by computing the z-score. Depending on the z-score, we bucket the attention in none (z < 0), low (0 < z < 1), medium (1 < z < 2), or high (2 < z). We plan to validate the efficacy of the attention weights with human users in future research.

Table 7: Best Hyperparameters settings for development set. R: Random, L: Linear-Normalized, S: Softmax-Normalized.

| | IEMOCAP | | | | SEMAINE | | | |
|---|---|---|---|---|---|---|---|---|
| | Classif. MA(4) | Val. MAE | Act. MAE | Dom. MAE | Act. MAE | Int. MAE | Pow. MAE | Val. MAE |
| Dev. set | 0.714 | 0.482 | 0.492 | 0.570 | 0.118 | 0.164 | 0.135 | 0.138 |
| Test set | 0.747 | 0.497 | 0.499 | 0.587 | 0.184 | 0.215 | 0.164 | 0.169 |
| Learning rate | 6.21 e-03 | 2.21 e-02 | 1.44 e-02 | 3.13 e-02 | 2.43 e-02 | 1.58 e-02 | 3.04 e-03 | 2.94 e-02 |
| Batch size | 43 | 43 | 62 | 77 | 41 | 43 | 33 | 62 |
| Pri GRU | 256 | 256 | 256 | 256 | 256 | 256 | 192 | 256 |
| Aux GRU | 1 | 1 | 1 | 1 | 1 | 1 | 65 | 1 |
| L2 regularization | 2.25 e-05 | 0 | 1.43 e-05 | 2.41 e-04 | 0 | 0 | 0 | 0 |
| Content-Size | 18 | 4 | 3 | 3 | 17 | 21 | 19 | 14 |
| GRU dropout | 0.27 | 0.49 | 0.33 | 0.30 | 0.06 | 0.17 | 0.05 | 0.07 |
| Recurrent dropout | 0.04 | 0.03 | 0.02 | 0.32 | 0.25 | 0.28 | 0.08 | 0.26 |
| Epoch No. | 232 | 161 | 74 | 14 | 48 | 116 | 110 | 110 |
| Aux. weights assignment | S | R | L | L | L | R | S | R |
| Main loss weight | 0.840 | 0.560 | 0.810 | 0.610 | 0.870 | 0.940 | 0.950 | 0.920 |
| AU05 loss weight | 0.009 | 0.013 | 0.014 | 0.000 | 0.000 | 0.002 | 0.002 | 0.008 |
| AU17 loss weight | 0.009 | 0.027 | 0.009 | 0.003 | 0.002 | 0.004 | 0.002 | 0.003 |
| AU20 loss weight | 0.009 | 0.028 | 0.0 | 0.001 | 0.000 | 0.007 | 0.002 | 0.006 |
| AU25 loss weight | 0.009 | 0.031 | 0.006 | 0.002 | 0.015 | 0.008 | 0.002 | 0.009 |
| Happy tone loss weight | 0.010 | 0.023 | 0.038 | 0.006 | 0.021 | 0.013 | 0.002 | 0.003 |
| Sad tone loss weight | 0.010 | 0.028 | 0.053 | 0.008 | 0.037 | 0.010 | 0.002 | 0.003 |
| Angry tone loss weight | 0.010 | 0.016 | 0.034 | 0.006 | 0.055 | 0.011 | 0.002 | 0.000 |
| Fear tone loss weight | 0.010 | 0.043 | 0.036 | 0.004 | 0.000 | 0.004 | 0.002 | 0.003 |
| $Y_{t-1}$ loss weight | 0.010 | 0.030 | 0.0 | 0.040 | 0.000 | 0.000 | 0.005 | 0.006 |
| $Y_{t-2}$ loss weight | 0.011 | 0.022 | 0.0 | 0.049 | 0.000 | 0.000 | 0.004 | 0.004 |
| $Y_{t-3}$ loss weight | 0.010 | 0.006 | 0.0 | 0.046 | 0.000 | 0.000 | 0.003 | 0.006 |
| $Y_{t-4}$ loss weight | 0.010 | 0.040 | 0.0 | 0.044 | 0.000 | 0.000 | 0.003 | 0.007 |
| $Y_{t+1}$ loss weight | 0.010 | 0.042 | 0.0 | 0.042 | 0.000 | 0.000 | 0.005 | 0.005 |
| $Y_{t+2}$ loss weight | 0.011 | 0.0 | 0.0 | 0.052 | 0.000 | 0.000 | 0.004 | 0.006 |
| $Y_{t+3}$ loss weight | 0.010 | 0.050 | 0.0 | 0.043 | 0.000 | 0.000 | 0.003 | 0.008 |
| $Y_{t+4}$ loss weight | 0.010 | 0.038 | 0.0 | 0.044 | 0.000 | 0.000 | 0.003 | 0.003 |

