# OpenReview forum: "Reusing Preprocessing Data as Auxiliary Supervision in Conversational Analysis"
_ICLR.cc/2021/Conference — Reject_

### Official Review · AnonReviewer4 · 2020-10-26
**Overall okay paper with some technical drawbacks**

**Rating:** 5
**Confidence:** 4

**Review:**

This paper tackles conversational analysis problem and more specifically the Primary Multi-Task Learning.

The three hypotheses the paper tries to address are interesting and important.

- Hypothesis H1: The introduced set of auxiliary supervision features improves primary MTL.

In this section, I am not quite sure if using future labels would cause information leakage to the primary task. If so, this would make the conclusion questionable.

- Hypothesis H2: When the primary branch is given maximum learning capacity, it would not be outperformed by models with primary branch having less than the maximum learning capacity.

In this section, I am not sure what is the fairest way to define learning capacity. Is increasing number of GRUs the best way? What if simpler model architecture can be used for auxiliary targets?

- Hypothesis H3: Auxiliary supervision at the lower hierarchy yields better primary MTL as compared to flat-MTL.
For this, where is the experimental result for using them as features instead of using as targets?

Overall, the paper adopts an experiments-driven approach to test the three hypotheses, but the main issue is that this approach adopts a specific neural network method. So how can we make sure that the conclusions always hold then?

Some minor observation: it seems the multi-modal approach in the CVPR 2020 paper "Multimodal Categorization of Crisis Events in Social Media" can be used here so that in Figure 3, concat can be used with better attention mechanisms.

---

> ### Author Response · Authors · 2020-11-22
> **Response to AnonReviewer4**
>
> Thank you for the insightful review. Please find below our response to each of the four points raised.
>
> > (1) In this section (H1), I am not quite sure if using future labels would cause information leakage to the primary task. If so, this would make the conclusion questionable.
>
> Response:
>
> We do not see how using future labels as supervisors would cause information leakage. We think that this is a problem only if we use the future labels as an input.
>
> We have added five citations that use predicting future labels as auxiliary tasks to help learning “Lastly, predicting future labels as auxiliary tasks can help in learning (Caruana et al., 1996; Cooper et al., 2005; Trinh et al., 2018; Zhu et al.,2020; Shen et al., 2020)” in the section “Hypothesis H1: The introduced set of auxiliary supervision features improves primary MTL.”
>
> Could you please elaborate on why using future labels as supervisors could cause information leakage?
>
>
> > (2) Hypothesis H2: When the primary branch is given maximum learning capacity, it would not be outperformed by models with primary branch having less than the maximum learning capacity. In this section, I am not sure what is the fairest way to define learning capacity. Is increasing number of GRUs the best way? What if simpler model architecture can be used for auxiliary targets?
>
>
> Response:
>
> In terms of fairness, we believe that changing the number of GRUs in the auxiliary and primary branch is fair because both branches are using the identical type of neural network. Increasing the number of GRUs in one branch would necessarily imply increasing the learning capacity relative to the other branch.
> After establishing that it is fair, we acknowledge that defining learning capacity as the number of GRUs can be seen as an over-simplification – for given the number of parameters, the learning capacity can differ between different types of network, say a single layer perceptron. While we agree that experimenting with other architectures in the auxiliary branch is an interesting research extension, we have added this as one of the limitations in section 6.
>
>
> > (3) Hypothesis H3: Auxiliary supervision at the lower hierarchy yields better primary MTL as compared to flat-MTL. For this, where is the experimental result for using them as features instead of using as targets?
>
> Response:
>
> We have constrained ourselves to use text-only inputs because we want to emulate how humans analyze conversations using Jefferson transcripts. The experiments are therefore set up to compare the different ways to use the auxiliary supervisors holding the text-only inputs constant. Therefore, this is beyond the scope of this paper.
>
> However, we predict that it would bring higher performance but at the qualitative cost of losing text-only inputs’ interpretability, which weakens our overarching narrative of creating interpretable (text-only) conversation analysis products. For quantitative comparison, we added the performance of the previous state-of-the-art approach (M3ER) in Table 4.
>
> Our approach is different from the existing state of the art approaches, where the multimodal numeric vectors are fused with numeric vectors from the text. We described the entire conversation in the text to facilitate ease of interpretability. This is elaborated in the new section (5.1). We also added the sentence, "Being fully text-based means that the analysis can be printed out on paper, without the need for speakers nor monitors to replay the conversation on a computer." in section 4.2. To demonstrate our HAN-ROCK model's interpretability appeal, we have visualized the difference in attention weights when Primary MTL is in effect vs. STL in Appendix A6.
>
> In addition, for some of the auxiliary supervisors, such as future and historical target labels, it is not possible to supply these as inputs (because these are supplied by humans) for a system that is deployed for real-time use-cases. We appreciate that the reviewer meant this for evaluation, but the evaluation from using historical/future human labels as inputs will be too optimistic because we would not have them during real-time deployment. Using these human labels as supervisors however, means that we would not require them during real-time deployment.
>
>
> > (4) Overall, the paper adopts an experiments-driven approach to test the three hypotheses, but the main issue is that this approach adopts a specific neural network method. So how can we make sure that the conclusions always hold then?
>
> Response:
>
> Agreed, this is a limitation despite our efforts to validate this across 8 tasks and 2 datasets. We picked HAN because of its ease of interpretability and hierarchical attention over the three tiers (words, talkturns, and tasks) of the internal representations. We added this limitation in section 6.

---

### Official Review · AnonReviewer3 · 2020-10-27
**The paper addresses multi-task learning for multimodal emotion recognition on two existing datasets (IEMOCAP and SEMAINE).**

**Rating:** 5
**Confidence:** 5

**Review:**

The paper addresses multi-task learning for multimodal emotion recognition on two existing datasets (IEMOCAP and SEMAINE).
Strengths:
*The issues addressed in this paper are very relevant. The use of a multi-task learning framework to tackle the lack of labeled data and the noisy labels for Affective computing research is a very interesting and still unexplored research line.

Weaknesses:
*This work is motivated by the analysis of video-conferencing videos which is indeed a crucial and topical issue. However, this motivation is a little bit heavy-handed as the processed data are very different from video-conferencing videos (face-to-face human interactions for IEMOCAP and human-agent/Woz interactions for SEMAINE database).
*Contrarily to what is claimed in the abstract, the relevance of the auxiliary task is not investigated in the paper. The first type of auxiliary tasks is predicting outputs using external predictors (Open face in order to obtain action units activations from computer vision and Vokaturi in order to extract emotion categories from the prosody). The primary tasks (4-classes emotion classification using the same 4 emotions, and dimensions prediction) are very close to these auxiliary tasks.  The second type of auxiliary tasks is predicting contexts such as done in language models and could be considered unsupervised pre-training objectives used for learning representations (but it is not described like that). Thus, if we push the reflection a little bit further, H1  seems to validate that using external off-the-shelf predictors -that are predicting the same outputs as the ones of the primary tasks- improves the results of the proposed predictor, which is not so interesting.
*The emotion categories that are considered are four of the Big-six of Ekman and I am wondering how relevant it is. For example, how often fear is observed in the SEMAINE and IEMOCAP datasets?
*The method used to provide a multimodal representation of the data for the inputs seems interesting (as I understand it: extracting multimodal features and generating an augmented text containing multimodal narratives using MONAH, a previous system proposed by the authors) but it's difficult to understand what this method brings compared to the state of the art of learning multimodal representations. Besides, it will be interesting to discuss shortly the performance of MONAH on the two datasets of human-human interactions (maybe showing some outputs).
*The proposed MTL framework relies on a combination of existing models (HAN and ROCK). Thus the contribution is rather experimental than methodological.
*Considering the problem of multi-task learning requires identifying and differentiating tasks and domains and the paper fails in doing this. In Section 5, 8 tasks are identified while 2 datasets and 5 or 6 tasks are actually considered (emotion classif, valence prediction,activation prediction, Dominance/Power prediction)


Typos :
*abstract: it may generalizes
*Section 3.2: indicateS
*Section 4.1: valence

---

> ### Author Response · Authors · 2020-11-22
> **Response to AnonReviewer3 (Part 1 of 2)**
>
> Thank you for the insightful review and listing the typos. Please find below our response to the first five of the ten points raised.
>
> > (1) This work is motivated by the analysis of video-conferencing videos which is indeed a crucial and topical issue. However, this motivation is a little bit heavy-handed as the processed data are very different from video-conferencing videos (face-to-face human interactions for IEMOCAP and human-agent/Woz interactions for SEMAINE database).
>
> Response:
>
> This is how people analyze conversations manually in the linguistics community. The Jefferson transcription is an existing, validated scientific instrument to analyze conversation.
> The motivation of our transcription system is to emulate the Jefferson transcription system, where conversation analysts manually describe the conversation in words and symbols, describing how the speech was articulated together with what was said. We have added the state-of-the-art methods for comparison in Table 4. This point is related to “*The method used to provide a multimodal representation of the data for the inputs seems interesting…”
>
>
> > (2) Contrarily to what is claimed in the abstract, the relevance of the auxiliary task is not investigated in the paper.
>
> Response:
>
> The relevance of the auxiliary tasks is investigated in the paper. Table 1 shows that the relevance of “ap” vs. “aphf” depends on the primary tasks. Perhaps the word relevance has a different connotation with the reviewer? To avoid confusion, we have changed all references of “relevance” auxiliary tasks to “beneficial” auxiliary tasks.
>
> > (3) The first type of auxiliary tasks is predicting outputs using external predictors (Open face in order to obtain action units activations from computer vision and Vokaturi in order to extract emotion categories from the prosody). The primary tasks (4-classes emotion classification using the same 4 emotions, and dimensions prediction) are very close to these auxiliary tasks.
>
> Response:
>
> The selected tasks are indeed closely related (but not identical) to the primary task. The reason we picked tasks that are closely related is to provide inductive bias from multiple sources of supervision whilst avoiding negative transfer.
> We have added the sentence, “The benefit of adding multiple tasks is to provide inductive bias through multiple noisy supervision (Caruana, 1997; Lipton et al., 2015; Ghosn & Bengio, 1997).”
>
>
> > (4) The second type of auxiliary tasks is predicting contexts such as done in language models and could be considered unsupervised pre-training objectives used for learning representations (but it is not described like that).
>
> Response:
>
> This is likely a misunderstanding. We clarified that in the paper and added the sentence “Although historical and future target labels can be used as a pre-training objective and fine-tuned on the current target label, sequential transfer learning is not the focus of this paper.” to the section “Hypothesis H1: The introduced set of auxiliary supervision features improves primary MTL”
> To AnonReviewer4’s point, we have also added more citations to show that training on future labels is existing practice.
> We disagree that it is “unsupervised”, because the historical and future (context) target labels are still provided by humans and not deducible from the input data. This is unlike, say, word2vec, where the pretraining is done on context words already present in the text data.
>
>
> > (5) Thus, if we push the reflection a little bit further, H1 seems to validate that using external off-the-shelf predictors -that are predicting the same outputs as the ones of the primary tasks- improves the results of the proposed predictor, which is not so interesting.
>
> Response:
>
> Again, we clarify that our paper is focused on multi-task learning, not single-task learning (STL), making this extrapolated reflection not grounded.
> Indeed, if we were using external off-the-shelf predictors that take in videos and predict the primary tasks, (1) we would have skipped the interpretable multimodal narrative creation step, making our approach not much different from the other existing state-of-the-art approach that fuses numeric vectors from different modalities; (2) We would have stacked models for different modalities together for STL. (3) The point about the inductive bias provided by multiple noisy supervisors onto randomly initialized weights becomes absent in this extrapolated STL view. And this is not what we are doing.
> To prevent this misunderstanding, we have added the sentence, “The benefit of adding multiple tasks is to provide inductive bias through multiple noisy supervision (Caruana, 1997; Lipton et al., 2015; Ghosn & Bengio, 1997)” to the related works section (2).

---

> ### Author Response · Authors · 2020-11-22
> **Response to AnonReviewer3 (Part 2 of 2)**
>
> Please find below the second five of the ten points raised.
>
> > (6) The emotion categories that are considered are four of the Big-six of Ekman and I am wondering how relevant it is. For example, how often fear is observed in the SEMAINE and IEMOCAP datasets?
>
> Response:
>
> The four-class emotion categories considered are in-line with other works on the IEMOCAP dataset. We have clarified this in section 4.1, with the sentence “The  four-class  emotions  classification  target  is  common  (Latif  et  al.,2020; Xia & Liu, 2015; Li et al., 2019; Hazarika et al., 2018b; Mittal et al., 2020), albeit not universal. Some researchers have gone up to five (Chang & Scherer, 2017) or six (Majumder et al., 2019; Hazarika et al., 2018a) or nine-class emotions classification (Zadeh et al., 2018a) target.”
>
> Fear is not a class label within IEMOCAP. The breakdown of the emotion counts from IEMOCAP is Sad (20%); Anger (20%); Happy (30%); Neutral (30%); The four tasks investigated in SEMAINE are all regression problems, i.e., no classification problem is presented.
>
>
> > (7) The method used to provide a multimodal representation of the data for the inputs seems interesting (as I understand it: extracting multimodal features and generating an augmented text containing multimodal narratives using MONAH, a previous system proposed by the authors) but it's difficult to understand what this method brings compared to the state of the art of learning multimodal representations.
>
> Response:
>
> This is how people analyze conversations manually. Jefferson transcription is an existing, validated scientific instrument to analyze conversation.
>
> This paper's contribution is not the creation of the MONAH transcripts, but how to more efficiently train on multimodal text-based inputs using preprocessed data as auxiliary supervisors. Therefore, our experiments are focused on improvements from using different supervisors (H1) and different model architectures to use the supervisors (H2 and H3), holding the input constant (text-only inputs).
>
> We agree that it is helpful to provide the state-of-the-art results for comparison, but we also believe that the constraint arising from using text-only inputs means that performance will not match state-of-the-art, which is confirmed in Table 4.
>
> Despite the constraint on performance, we believe this line of research of using text-only inputs to analyze conversations is a step towards augmenting how humans have been analyzing and annotating conversations in the linguistics community, imitating the use of the Jefferson transcription system.
>
> We have added a new section (5.1) to elaborate on the above.
>
>
> > (8) Besides, it will be interesting to discuss shortly the performance of MONAH on the two datasets of human-human interactions (maybe showing some outputs).
>
> Response:
>
> We think it is a great suggestion to show some outputs because this also helps motivate the overarching motivation of using text-only inputs and the appeal of HAN’s interpretability on text classification.
>
> We have visualized the difference in attention weights when Primary MTL is in effect vs. STL in Appendix A6. We plan to validate the efficacy of the visualizations with human users in future research.
>
>
> > (9) The proposed MTL framework relies on a combination of existing models (HAN and ROCK). Thus the contribution is rather experimental than methodological.
>
> Response:
>
> Our proposed methodology successfully fused the interpretable HAN, with the ROCK which was invented for the purposes of computer vision. The HAN-ROCK architecture enables analysts to bring in auxiliary tasks to improve on the primary task learning and retain its interpretability over words and talkturns. Our results confirm all three hypotheses which contributes new knowledge, and it has been tested across 8 primary tasks on 2 datasets which suggests generalizability (at least in the domain of conversational analysis).
>
> > (10) Considering the problem of multi-task learning requires identifying and differentiating tasks and domains and the paper fails in doing this. In Section 5, 8 tasks are identified while 2 datasets and 5 or 6 tasks are actually considered (emotion classif, valence prediction, activation prediction, Dominance/Power prediction)
>
> Response:
>
> We agree that because the task names are similar across the two datasets, they could be paired together and seen as similar tasks across the two domains. However, cross-domain learning, neither single-task nor multi-task, is scoped in this paper.
>
> We clarified and added, “We note that the valence, power, and activation tasks might be related across the two datasets, but cross-domain learning is beyond the scope of this paper.” to section 4.1.
>
> Instead, the scope of this paper is how to train on one dataset more efficiently by reusing preprocessed multimodal data from the same dataset as auxiliary supervision.

---

### Official Review · AnonReviewer1 · 2020-10-28
**An interesting work on multi-task learning but novelty is limited**

**Rating:** 6
**Confidence:** 4

**Review:**

This paper addresses challenges faced in the multi-task learning (MTL) models used in analyzing multimodal conversational data. The main challenge paper is trying to solve is on how to select relevant auxiliary tasks that avoid negative transfer. The authors explore how the preprocessed data used for feature engineering can be re-used as auxiliary tasks in the model. The authors identified sixteen relevant auxiliary tasks, identified a method to distribute learning capacity between primary and auxiliary tasks and proposed a relative supervision hierarchy between primary and auxiliary tasks. An extensive set of experiments are conducted to show the effectiveness of the approach.

The paper is trying to tackle an important problem faced by multi-task learning (MTL) and the way different aspects of the problem are explained in paper is valuable. The paper takes a systematic approach to address various aspects and conduct extensive experiments to show that having auxiliary task in the model significantly improved performance of the primary task, assigning higher learning capacity (number of parameters) to primary tasks helps in the performance and placing auxiliary supervision at the lower hierarchy significantly improves the performance on primary task. The paper is well written and easy to follow.

Although the paper is trying to tackle an important challenge, the novelty or contribution of the paper is limited. The useful ideas concluded by the paper has been previously identified in the community, for example, NLP community has been using auxiliary tasks like NER to improve primary task’s performance. Also, hierarchy in which to structure primary and auxiliary tasks in a model is also somewhat discussed previously. Another shortcoming of the paper is that the proposed solutions are evaluated on a specific domain and it is not clear if these findings are general enough to be applied to other domains.

---

> ### Author Response · Authors · 2020-11-22
> **Response to AnonReviewer1**
>
> Thank you for the insightful review. Please find below our response to each of the three points raised.
>
> > (1) Although the paper is trying to tackle an important challenge, the novelty or contribution of the paper is limited. The useful ideas concluded by the paper has been previously identified in the community, for example, NLP community has been using auxiliary tasks like NER to improve primary task’s performance.
>
> Response:
>
> We added two new paragraphs in the literature review section (2) to position the contribution of our paper amongst the NLP community and multimodal conversational analysis. The research gap we are highlighting is that auxiliary supervisors are not used in multimodal analysis, and hence we identify the sixteen auxiliary supervisors that we found to be useful. The sixteen auxiliary supervisors belong to two types: (a) future and past labels (context), and (b) granular percentile prediction of video and audio attributes. As pointed out by AnonReviewer2, these two sources of information have already been used by existing state-of-the-art multimodal models as inputs. The main difference in our approach is that we show how to use them as both inputs and supervisors (outputs) to improve performance on the primary task.
>
> > (2) Also, hierarchy in which to structure primary and auxiliary tasks in a model is also somewhat discussed previously.
>
> Response:
>
> Yes, our approach of structuring primary tasks downstream of auxiliary tasks is heavily influenced by the existing ROCK architecture (Mordan et al., 2018).
> Our proposed methodology successfully fused the interpretable HAN, with the ROCK which was invented for the purposes of computer vision. The HAN-ROCK architecture enables analysts to bring in auxiliary tasks to improve on the primary task learning and retain its interpretability over words and talkturns. Our results confirm all three hypotheses which contributes new knowledge, and it has been tested across 8 primary tasks on 2 datasets which suggests generalizability (at least in the domain of conversational analysis).
> To demonstrate the interpretability appeal of our HAN-ROCK model, we have visualized the difference in attention weights when Primary MTL is in effect vs. STL in Appendix A6.
>
> > (3) Another shortcoming of the paper is that the proposed solutions are evaluated on a specific domain and it is not clear if these findings are general enough to be applied to other domains.
>
> Response:
>
> We evaluated our proposed solutions on eight tasks across two conversational analysis datasets. While these are quite different, we agree that future work should expand this evaluation to more domains, but do not think this could be scoped in this paper. We added this limitation in section 6.

---

### Official Review · AnonReviewer2 · 2020-10-31
**Official Blind Review #2**

**Rating:** 6
**Confidence:** 4

**Review:**

This paper studies how the preprocessed data can be reused as auxiliary tasks in primary multi-task learning (MTL) for the multimodal emotion detection task. The authors propose and test three hypotheses for primary MTL. Two different hierarchical-level models, FLAT-MTL hierarchical attention model and HAN-Rock model, are proposed to improve the performance of the primary MTL.

Overall, the definition of the task and the overall architecture of this paper are both clear and straightforward, making this paper easy to understand. To explore how to use the preprocessed data as auxiliary tasks in primary MTL, the authors present a nicely executed study and test three hypotheses. Besides, the proposed model achieves impressive results on two datasets in terms of the multimodal emotion detection task.

Here are some of my questions and concerns for the paper:
1) The contribution of this paper seems to be limited since the idea of using visual features, audio features, and context as auxiliary information is similar to lots of emotional classification models such as ICON, CMN. In addition, the proposed model for MTL is also straightforward.
2) More related works about multi-task learning and multimodal emotion detection tasks should be included.
3) More in-depth experimental analysis should be carried out. For example, what is the individual performance of models on four different emotions under the three hypotheses?
4) Some symbols in formulas lack explanations. For instance, X_t,X_(t-1),X_(t-2) on page 2 and b_w,W_w,u_w,u_s on page 3 are all not clearly defined.

---

> ### Author Response · Authors · 2020-11-22
> **Response to AnonReviewer2**
>
> Thank you for the insightful review. Please find below our response to each of the four points raised.
>
> > (1) The contribution of this paper seems to be limited since the idea of using visual features, audio features, and context as auxiliary information is similar to lots of emotional classification models such as ICON, CMN. In addition, the proposed model for MTL is also straightforward.
>
> Response:
>
> Although existing emotional classification models do use context, visual and audio features as inputs, we would like to point out two important differences. Our first difference in approach is that because we want to emulate how humans are doing manual conversational analysis, we constrained our inputs to be text-only, but this text-only input is annotated with multimodal cues. Our second difference in approach is that the existing approaches (ICON, CMN, MFN, DialogueRNN, M3ER) use the multimodal information as inputs; we are using them as inputs and supervision. We have added a paragraph to clarify these two differences in the related works, section 2.
>
> > (2) More related works about multi-task learning and multimodal emotion detection tasks should be included.
>
> Response:
>
> We have expanded section 2 (related works) substantially with two paragraphs, one for multi-task learning (specifically in the area of NLP) and one for multimodal emotion detection. With the two new added paragraphs, we clarified the contribution of this work relative to the newly added related works.
>
> In summary, “Our approach is different in two ways.
>
> (1)	Our audio and video information is encoded within text before feeding only the text as input. Having only text as input has the benefits of interpretability, and the ability to present the conversational analysis on paper. This is similar to how the linguistics community performs manual conversational analysis using the Jefferson transcription system, where the transcripts are marked up with symbols indicating how the speech was articulated.
>
> (2)	Instead of using the audio and video information as only inputs to a Single Task Learning (STL) model, the contribution of this paper is that we demonstrate how to use multimodal information in both input and as auxiliary supervisors to provide inductive bias that helps the primary task.”
>
> > (3) More in-depth experimental analysis should be carried out. For example, what is the individual performance of models on four different emotions under the three hypotheses?
>
> Response:
>
> That is a great idea. We have shifted Table 4 into a new subheading, section 5.1, where we elaborated on the individual performance of models on four different emotions under the three hypotheses. This exercise has also helped us discover a new finding under H2. We observed significant performance degradation for all classes, moving from assigning 256 to the primary branch, to assigning 1 to the primary branch.
>
> > (4) Some symbols in formulas lack explanations. For instance, X_t,X_(t-1),X_(t-2) on page 2 and b_w,W_w,u_w,u_s on page 3 are all not clearly defined.
>
> Response:
>
> Upon feedback, we have crosschecked every symbol in formula is explained in prose, and removed all unnecessary symbols X_t,X_(t-1),X_(t-2), Y_t etc.

---

### Decision · Program_Chairs · 2021-01-07
**Final Decision**

**Decision:**

Reject

**Comment:**

The initial reviews for this paper were very borderline. The authors provided detailed responses as well as a few additional results and observations. The authors' responses answered the reviewers' questions and addressed their main comments (including in the discussion of related works as well as with more in-depth analysis in a new Section 5.1). Unfortunately, the reviewers did not come to a consensus.

Overall, this paper extends some current methodology for emotional classification, is well-executed, and provides a reasonably thorough study. The results are somewhat in line with previous results from other fields (and notably NLP), but the authors demonstrate the efficacy of using primary multi-task learning for multimodal conversational analysis.

Unfortunately, this paper also has some flaws as highlighted by the initial reviews. As stated above, the authors did provide a strong rebuttal, but given the different comments raised by the reviewers that spanned many aspects of the paper including motivation, possibly limited contribution and novelty, missing related work, somewhat shallow analysis of the results, I find that another full round of reviewing would be useful to assess the paper.

As a result, this remains a very borderline paper, and given the strong competition at this year's conference, I cannot recommend acceptance at this stage.

I suggest that the authors incorporate some of the discussions from this forum (and especially with respect to related work, new findings, and clearly defining the motivation and contribution of this work) into the next version of their paper.